# In Silico Predicting the Presence of the S100B Motif in Edible Plants and Detecting Its Immunoreactive Materials: Perspectives for Functional Foods, Dietary Supplements and Phytotherapies

**DOI:** 10.3390/ijms25189813

**Published:** 2024-09-11

**Authors:** Vincenzo Romano Spica, Veronica Volpini, Federica Valeriani, Giovanni Carotenuto, Manuel Arcieri, Serena Platania, Tiziana Castrignanò, Maria Elisabetta Clementi, Fabrizio Michetti

**Affiliations:** 1Department of Movement, Human and Health Sciences, University of Rome “Foro Italico”, 00135 Rome, Italy; v.volpini@studenti.uniroma4.it (V.V.); federica.valeriani@uniroma4.it (F.V.); serena.platania@hotmail.com (S.P.); fabriziomichetti.office@gmail.com (F.M.); 2Department of Ecological and Biological Sciences, University of Tuscia, Viale dell’Università s.n.c., 01100 Viterbo, Italy; giovanni.carotenuto@studenti.unitus.it (G.C.); tiziana.castrignano@unitus.it (T.C.); 3Department of Health Technology, Technical University of Denmark, 2800 Kongens Lyngby, Denmark; s230158@dtu.dk; 4Genes, Via Venti Settembre 118, 00187 Roma, Italy; 5Istituto di Scienze e Tecnologie Chimiche “Giulio Natta” SCITEC-CNR, L.go F. Vito 1, 00168 Rome, Italy; elisabetta.clementi@scitec.cnr.it; 6Department of Medicine, LUM University, 70010 Casamassima, Italy

**Keywords:** S100B, plant, nutraceuticals, microbiota, diet, protein domain

## Abstract

The protein S100B is a part of the S100 protein family, which consists of at least 25 calcium-binding proteins. S100B is highly conserved across different species, supporting important biological functions. The protein was shown to play a role in gut microbiota eubiosis and is secreted in human breast milk, suggesting a physiological trophic function in newborn development. This study explores the possible presence of the S100B motif in plant genomes, and of S100B-like immunoreactive material in different plant extracts, opening up potential botanical uses for dietary supplementation. To explore the presence of the S100B motif in plants, a bioinformatic workflow was used. In addition, the immunoreactivity of S100B from vegetable and fruit samples was tested using an ELISA assay. The S100B motif was expected in silico in the genome of different edible plants belonging to the Viridiplantae clade, such as *Durio zibethinus* or *Malus domestica* and other medicinal species. S100B-like immunoreactive material was also detected in samples from fruits or leaves. The finding of S100B-like molecules in plants sheds new light on their role in phylogenesis and in the food chain. This study lays the foundation to elucidate the possible beneficial effects of plants or derivatives containing the S100B-like principle and their potential use in nutraceuticals.

## 1. Introduction

S100B belongs to the S100 protein family which includes at least 25 calcium-binding proteins [1]. These molecules are mostly homodimers that share similar structures. Members of the S100 protein family share a high degree of structural similarity, despite having only 25–65% similarity in amino acid sequences. Each S100 protein has two Ca^2+^- modulated motifs interconnected by a hinge region, each resulting in a helix-loop-helix arrangement [2,3]. The calcium-binding site belongs to the EF-hand motif, i.e., a pentagonal structure that loops the calcium ion [4]. The C-terminal EF-hand binding loop is a typical EF-hand motif, consisting of 12 residues, whereas the N-terminal “pseudo-canonical” EF-hand Ca^2+^-binding loop, known as the variable EF-hand motif, with 14 residues, is a distinguishing feature of the S100 family. The typical S100 protein can have a quaternary structure as a symmetric dimer, with each monomer containing two EF-hand motifs. In particular, the protein S100B is made up of 81 *amino acids* organized in two functional domains, namely, the S100 domain (AA 4–46) and the EF-hand domain (residues 53–81). Its three-dimensional structure resembles a “knock-fist” configuration where the specific aminoacidic sequence of the S100 domain is characterized by an almost polar structure in the “fist” whereas the “knock” is mainly hydrophobic (Figure 1 and Appendix A).

S100B protein, which was the first S100 protein to be discovered [5], is concentrated in astrocytes in the nervous system, where it is also located in oligodendrocytes, Schwann cells, ependymal cells, retinal Müller cells, enteric glial cells and some neuron subpopulations, but is also present in extra-neural cell types, such as adipocytes, chondrocytes, melanocytes, Langerhans cells, dendritic cells of lymphoid organs, some lymphocyte cell types, adrenal medulla satellite cells, skeletal muscle satellite cells, tubular kidney cells, Leydig cells, and non-nervous structures of the eye [6,7,8,9,10,11]. As for other members of the S100 family, the S100B conformation and amino acid composition is highly conserved during phylogeny, suggesting that it may have critically conserved biological roles [12,13]. However, although many hypotheses have been formulated for its function(s), the role(s) of this protein remain(s) unclear [14]. S100B is currently regarded to act in a different manner depending on its local concentration in different tissues: trophic at physiological nanomolar concentrations, or toxic at higher micromolar concentrations [6,15]. Interestingly, the S100B protein has also been found in human breast milk, suggesting a role as a physiological trophic factor, possibly useful in newborn development, and with a potential role in diet [16,17,18]. S100B protein has also been detected in milk samples from cows, sheep, goats, and donkeys, with higher concentrations in cow and donkey samples although lower than in humans [19]. These findings open the possibility of considering S100B as a useful nutrient or a potential diet supplement. This consideration appears to be more realistic nowadays because the protein has recently been shown to actively interact in silico and in vivo with human gut microbiota [20,21]. The “knock-fist” three-dimensional structure is a conceptual prerequisite for an interaction involving S100B and other molecules, including the gut microbiota proteome. 

A systematic phylogenetic study of S100B in plants has never been performed. In light of the above indicated recent views, suggesting S100B to be a potential nutrient and also hinting at its possible interaction with microbiota, its presence in the plant kingdom also appears to ignite a novel interest in phytotherapeutic applications. This study explores the presence of the S100B motif in plant genomes, and of S100B-*like* immunoreactive material in different plant extracts, thus fostering the protein as a dietary supplement or active component in functional foods and a novel principle for phytotherapies. The results will open new perspectives on vegetable sources of S100B and expand knowledge of this still challenging protein expressed in different human tissues as well as in plants.

## 2. Results

### 2.1. In Silico Molecular Investigations

The S100B protein motif was expected in silico in different plant species. The root-mean-square deviation (RMSD) serves as a metric to quantify the average spatial separation between the atoms of the examined protein and the reference S100B structure. A diminished RMSD value correlates with heightened structural similarity, as elucidated in the tabulated results. Comprehensive protein assessments already yield acceptable outcomes; however, a more refined analysis focusing solely on conserved residues revealed an additional reduction in the RMSD (see the RMSD conserved column in Table 1 and Appendix A).

This observation underscores the notably elevated degree of structural homology within the S100B motif of the investigated proteins. Some examples of the plant homolog proteins aligned with the conserved S100B are reported in Figure 2 and Appendix A, showing the three-dimensional structural correspondence.

However, from a purely phylogenetic point of view, the S100B protein alone is not informative enough to clearly define the evolutionary history of green plants, but it can be noticeably regarded as an interesting biomarker. Afterwards, the conserved domain demonstrating significant structural homology with the benchmark protein S100B was identified and isolated. The obtained structures were aligned with the benchmark protein structure, and RMSDs and TM-align were computed to evaluate the precision of the structural similarity between the analyzed peptides and the reference protein [23]. The RMSD and TM-align calculations were normalized with respect to the reference structure of S100B. Finally, a satisfying alignment of both peptides analyzed with S100B was clearly observed. Notably, the comprehensive analysis of the entire protein structures had already revealed substantial similarities. However, a more discerning examination, isolating and comparing the S100B motif, elucidated an augmented level of similarity when contrasted with the reference protein. The results of the comparative structural analysis indicated that the three-dimensional structural similarity is significantly improved when only the conserved domain is considered. This observation supports the hypothesis that proteins have retained this biological function throughout evolution in the plant kingdom. These findings support the presence of proteins with a conserved S100B motif in different plants belonging to the plantae kingdom.

### 2.2. A S100B Dimeric EF-Hand Organization in Plants

To study the S100B protein among green plants, we looked for similar proteomic sequences from species belonging to the Viridiplantae clade, showing the highest homology with the human S100B motif (Figure 3).

We retained the top 100 organisms with the closest similarity to the human S100B protein. The individual proteomes were used to infer a phylogenetic tree, aiming to obtain an evolutionary history close to current taxonomic beliefs. Considering the limited size of the S100B protein, it managed to derive a satisfying phylogeny. Species with a close common ancestor were correctly clustered in the same groups, e.g., *Musa balbisiana* and *Ensete ventricosum*, the Gossypium genus, *Hibiscus syriacus*, and *Durio zibethinus*, the *Brassica genus*, *Eutrema salsugineum*, and *Raphanus sativus* (Figure 4). 

### 2.3. Immunodetection of S100B-like Material in Plant Extracts

In the same species where S100B motif was identified in silico, S100B-like immunoreactive material was also detected in plant samples. A total of 50% of tested samples were positive (*n* = 11/22) (Table 2). Among the powder samples, 50% were positive (*n* = 5/10), and 33.3% (*n* = 3/9) of fresh samples were positive. Regarding fruits, 77.8% of them were positive (*n* = 7/9), but all the tested vegetables were negative. Interestingly, aromatic plants (sage and laurel) were positive. Only açai samples were positive among berries. Notably, two members of the same family (Malvaceae), durian and baobab fruit, were positive for S100B-*like* immunoreactive material among the samples that tested positive. 

Figure 5 shows the standard curve of a representative experiment also indicating the location of durian fruit extracts regarding the concentration of S100B-like material. 

Immunodetection of S100B epitopes extracts from different plants supports the in silico findings, further confirming the presence of this molecular structure in selected plants.

## 3. Discussion

This study shows that the S100B motif appears *to be very* conserved in plants, and S100B-*like* immunoreactive materials are also detectable in plant species. In light of the pivotal importance and wide diffusion of calcium signaling in animal and vegetal kingdoms, the detection in plants of the protein domain characterizing a calcium-binding protein, such as S100B protein, is not surprising. In fact, in addition to calmodulin, which is probably the best-studied example of the EF-hand Ca^2+^-binding proteins, calcineurin and other novel calcium-dependent protein kinases have also been detected in plants [4]. Interestingly, the observation that the three-dimensional structural similarity is significantly improved when only conserved domains are considered, might underline the hypothesis that proteins have retained their biological functions throughout evolution in the plant kingdom. However, to the best of our knowledge, proteins of the S100 family as natural constituents of plants have not been reported, and this protein family is currently believed to be expressed only in vertebrates, where it is widely distributed among animal species and regarded to play an important, although not yet completely clarified, physiological or physio-pathological role [1,3,24]. Their presence in plants opens new perspectives to obtain sources of S100B active principle(s) from different plants and/or some of their derivatives potentially of interest for new research lines in nutrition and/or phytotherapy. The consistent detection of “phyto-protein” molecular structures displaying a S100B motif seems to be enticing and reasonable. Although with their own peculiarities, plants are regarded essentially to perform all those functions previously described in human and animal cells [25,26,27,28]. Among S100 proteins, this work also shows experimentally that the same definite plant species which were observed to display the S100 motif in silico, actually contain S100B-like immunoreactive material. Of course, it should be noted that these findings do not allow us to determine that these plants contain molecules that functionally overlap with animal S100B protein. Moreover, a larger number of different antibodies would significantly strengthen the observed data. In addition, antibodies specifically binding vegetable S100B are not available, as this study is the first to definitively address this topic. Thus, we have to define the molecule(s) reacting with the antibodies currently available as “S100B-like” immunoreactive material. However, the present experimental data, together with the in silico finding of a conserved S100B protein domain, allow us to infer that the molecule(s) present in these plants might exhibit structural and therefore functional characteristics that reasonably do not diverge importantly from those considered in animal tissues. In mammalians, S100B protein is known to exert a trophic beneficial effect at low physiological concentration and a pathogenic effect, behaving as a damage/danger-associated molecular pattern protein at a high concentration. Thus, also for plants, we might presume that at low physiological concentration (nM in animals), the S100B-like molecule(s) could exert a trophic role (the so-called “Jeckyll side”) and at a high pathological concentration (mM in animals), the molecule(s) could even play a toxic role (the so-called “Hyde side”). This may suggest both a role as a therapeutic principle and a role as a possible therapeutic target [6,7,13] for the protein. This unexpected perspective might open new avenues for both S100B studies and plant physiology. In the present state of knowledge, further studies will be needed to verify this intriguing possibility. Interestingly, these data, if conclusively delineated, would again propose phyto-S100B as a potential trophic factor available in aliments and food chains, in addition to its finding in mammalian milk [17,29]. Therefore, in addition to the selected edible plants reported here, other species may be revealed as a possible source for this protein principle in the future. Furthermore, following similarities between S100B in animals and plants, the possibility that the protein may also be secreted in plants, within microvesicles, cannot be ruled out, thus opening a novel perspective on its participation in nutritional and therapeutic effects of plant-derived microvesicles [30]. S100B-like immunoreactive material was found in different vegetables and fruits, such as aromatic plants, sage, and laurel, but also açai, durian, and baobab fruits from the Malvaceae family. Consuming these fruits and vegetables has been shown to have a significant positive impact on the health of the gastrointestinal tract and the overall well-being of human subjects [31]. Indeed, medicinal herbs, such as aromatic plants and sage, are rich in phytochemicals and their extracts are already added to functional food products as diet supplements, making them desirable ingredients also for functional beverages to improve gut microbiota and overall health [32,33]. The same can be observed for several vegetables and fruits included in the list of samples that are immunoreactive for S100B. Brazilian native fruits, such as açaí (*Euterpe oleracea* Mart.), contain bioactive compounds investigated for their ability to modulate intestinal microbiota [34] by decreasing the level of a genus, described as a hostile microorganism to intestinal microbiota, as some genera of Clostridium [35]. In addition, studies have shown that Baobab fruits (*Adasonia digitata*) have several phytochemical and biological activities. Specifically, the pulp powder has been found to stimulate the growth and/or activity of certain bacteria that are already present in the colon, increasing the biodiversity indices [36]. Recently, a study showed that *Durio zibethinus* was correlated with a reduction in pathogenic bacteria such as *Desulfovibrio* and an increase in the relative abundance of beneficial bacteria such as *Lachnospiraceae* in animal models [37]. Some molecules from *Artocarpus heterophyllus* Lam (Jackfruit) exhibited immune-stimulating activity in a mouse model, as well as changes to the microbiota [38,39]. Despite scientific studies revealing that vegetables and fruits have a wide diversity of bioactive compounds with beneficial effects on health, the mechanisms of their beneficial actions are not yet fully understood [40]. The finding of molecules at least similar to S100B in plants, in the light of the possible roles of the protein as a natural trophic factor in milk [17,29] and as an effector on microbiota biodiversity [20,21] might participate in the explanation of these phenomena, although, additional data will unravel mechanisms and detail the S100B beneficial effects by its assumption thought plants or their derivatives containing the S100B principle. 

## 4. Materials and Methods

### 4.1. In Silico Molecular Investigations

#### 4.1.1. Data Preparation

The human S100B protein was compared with similar compounds by running blastp against the UniProtKB and the SwissProt databases [41] (accessed in February 2024). Only organisms belonging to the Viridiplantae clade were retained. The results were manually filtered to remove duplicates and low-quality items. The best 100 proteomes were selected and the corresponding FASTA sequences were downloaded from UniProt [42]. Subsequently, they were aligned using MAFFT [43] (version 7.490) and trimmed with trimAl (version 1.4) [44]. Finally, IQ-TREE (version 2.2.2.6) [45] was employed to infer the final phylogenetic tree, using the Le and Gascuel substitution matrix [46] with a 4-category discrete gamma model (LG+G4) [47]. The results were compared with the phylogenetic tree built using the NCBI Common Taxonomy Tree [48] tool passing the Taxa IDs of the species we previously selected [49].

#### 4.1.2. Molecular Modelling

To predict the three-dimensional structure of the proteins that manifested better sequence homology, we employed the protein modelling software AlphaFold2 (version v2.3.0) [50]. The alignments required for this procedure were generated using MMseqs2 (version 2) [51] and HHsearch (version 1.12) [52] software, which allowed us to obtain detailed sequence relations between the proteins of interest. 

The outputs generated from the structural prediction were used to calculate the root mean square deviation (RMSD) and TM-align [22,23]. Subsequently, the structures were incorporated into PyMOL to calculate the solvent accessible surface area (SASA) based on the original structures of the proteins [53,54]. TM-align is an algorithm used for comparing sequence independent protein structures. When confronted with two protein structures of unknown equivalence, TM-align undertakes the generation of an optimized residual-residual alignment through dynamic programming heuristic iterations, grounded in structural similarity. The algorithm subsequently furnishes an optimal superimposition of the two structures, accompanied by a TM-score value that quantifies the degree of structural resemblance. The TM-score, constrained within the range of (0, 1), attains a value of 1 for a perfect structural match. In accordance with established structural statistics from the Protein Data Bank, TM-scores falling below 0.2 are indicative of randomly selected, unrelated proteins, whereas scores surpassing 0.5 generally signify congruent folding in SCOP/CATH.

### 4.2. In Field Investigations

#### 4.2.1. Sample Preparation

We used different ELISA kits from different manufacturers, using different antibodies (Abcam GR3360381-1, Cambridge, UK, and Millipore Merck EZHS100B-33K, Billerica, MA, USA) to verify the presence of S100B-like immunoreactive material in our plant extracts. Notably, the antibodies used, according to the indications offered by the manufacturers, reacted specifically with the beta subunit of the protein as present in humans, but also in other mammalian species, such as rat and mouse (as indicated for both antibodies) and hamster, ox, horse, monkey, pig, rabbit, (as indicated for Merck). No cross reactivity with other members of the S100 protein family (namely, S100A1, S100A6, S100A13) was observed using the Merck antibody, as tested by the manufacturer. The sensitivities declared by the manufacturers were 2.7 (Merck) and 139 pg/mL (Abcam), respectively. The samples used for these experiments were in fresh or dry form (Appendix A). Approximately 80–120 mg of vegetal organisms was weighted: the initial weights were gradually reduced because we experimentally noticed an optimal extraction efficiency with the lowest quantity of material. Into every sample, 1 mL of Cell Extraction Buffer PTR 1X (Abcam, 6300003, lot R1874, Amsterdam, The Netherlands), glass beads (Sigma-Aldrich, G1145-10G, lot 019K5306, Saint Louis, MO, USA) and Protease inhibitor 1:100 (Sigma-Aldrich, P2714-1BTL, lot 0000116640, Saint Louis, MO, USA) were added. To obtain a homogenate, samples were vortexed briefly and then incubated on ice for 20 min. After this time, they were centrifuged at 16,300× *g* for 20 min and the supernatants were transferred into clean tubes. The samples were then immediately assayed.

#### 4.2.2. Enzyme-Linked Immunosorbent Assay for S100B

An enzyme-linked immunosorbent assay (ELISA) for S100B (Abcam, ab234573, Amsterdam, the Netherlands and Millipore Merck EZHS100B-33K, Billerica, MA, USA) was carried out on the supernatants on duplicates according to the manufacturer’s protocol. Briefly, after the reagents, samples, and standards preparation, 50 µL of sample or standard were added to appropriate wells: both standards and samples were assayed in duplicate. A total of 50 µL of antibody cocktail (S100B Capture Antibody catalogue n. 630447, lot Q4137, S100B Detector Antibody catalogue n. 6304493 lot Q4138 and Antibody Diluent 4BI catalogue n. 6301010, lot Q3223) was added to all wells and left for incubation at room temperature for 1 h. In the washing process, supernatant was removed and 350 µL of Wash Buffer PT 1X was added; the process was repeated three times. Subsequently, 100 µL of substrate solution were added to each well and incubated for 5 min at room temperature; then, 100 more µL of Stop solution was added. Absorbance was measured on a microtiter plate reader (Bio Rad, iMark Microplate Absorbance Reader, Hercules, CA, USA) at 450 nm. The concentration of S100B (ng/mL) in the samples was interpolated from the standard curve obtained by the standards.

## Figures and Tables

**Figure 1 ijms-25-09813-f001:**
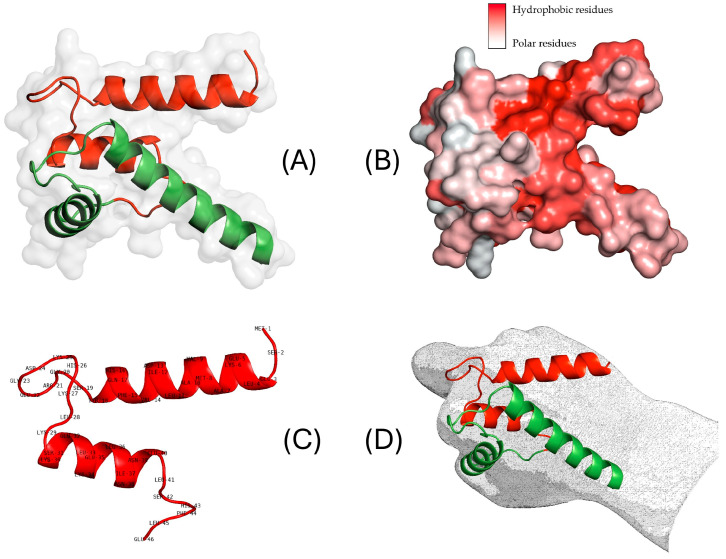
Visualization of S100B protein with the “S100 domain” highlighted in red and the “EF-hand domain” in green (**A**). Representation of the hydrophobic regions of S100B scaled from white (polar) to red (hydrophobic) (**B**), with the visualization of the aminoacidic sequence of the S100 domain (**C**). The three-dimensional “knock-fist“ structure of S100B (**D**).

**Figure 2 ijms-25-09813-f002:**
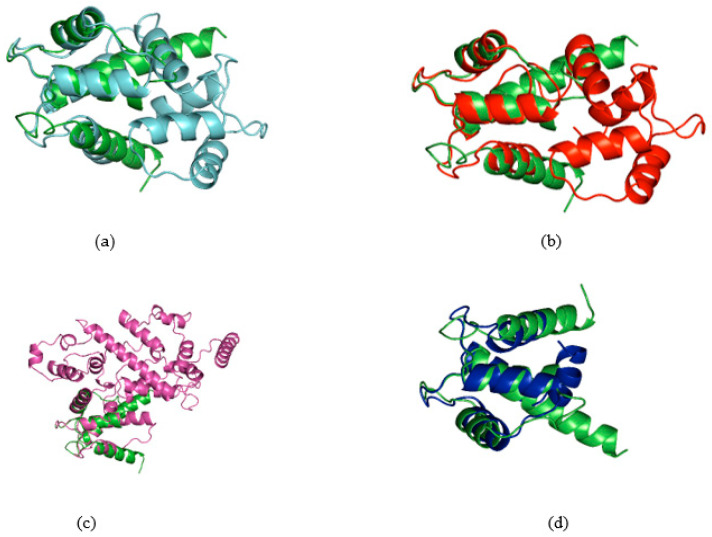
Alignments of conserved domains of *Durio zibethinus* in light blue (**a**), *Hibiscus syriacus* in red (**b**), *Malus domestica* in violet (**c**) and *Musa acuminata* subsp. malaccens in blue (**d**) with S100B protein in green.

**Figure 3 ijms-25-09813-f003:**
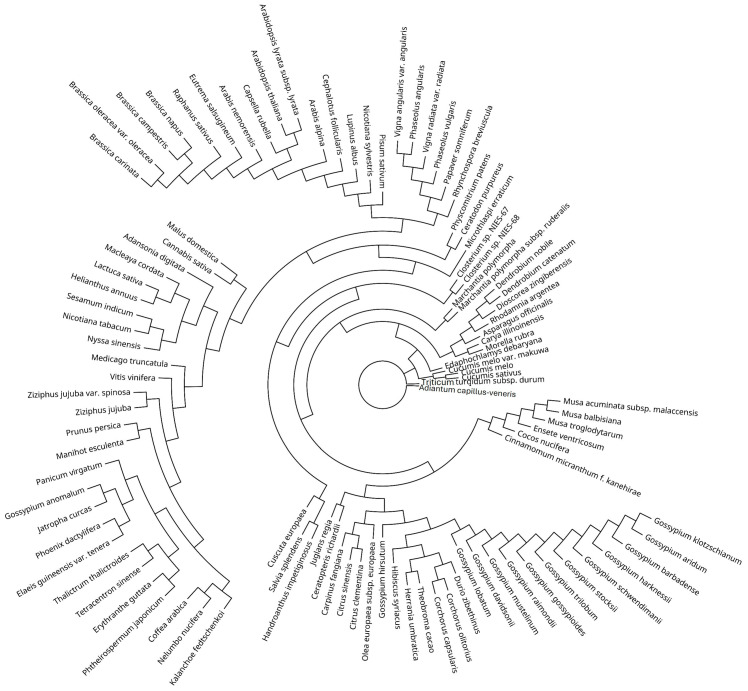
S100B-derived phylogenetic graphic.

**Figure 4 ijms-25-09813-f004:**
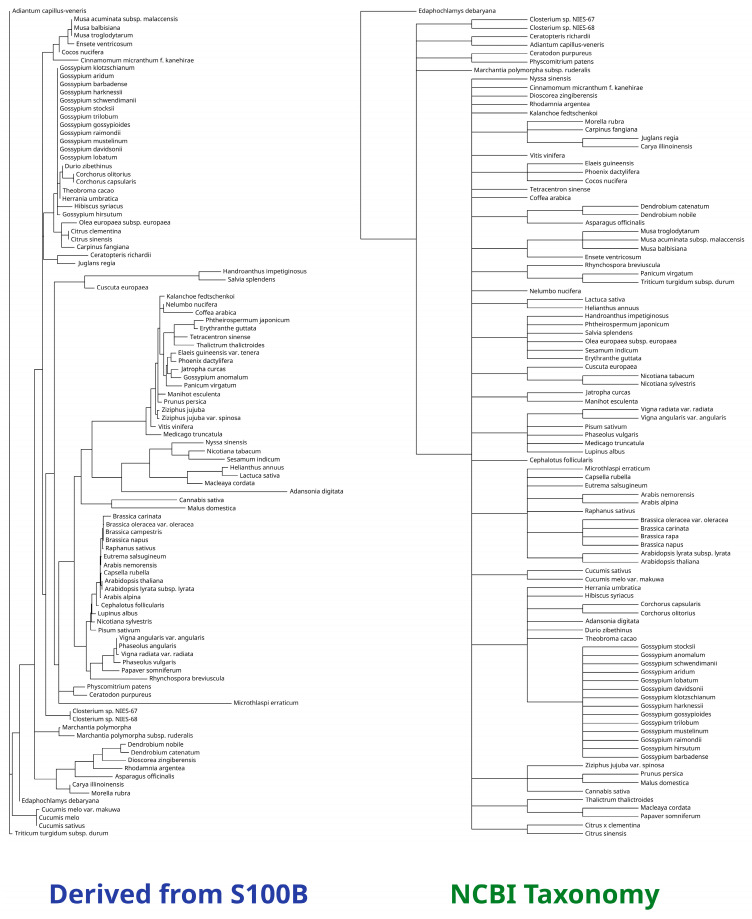
Comparison between the S100B phylogenetic tree and the NCBI taxonomy tree. The phylogenetic tree derived from S100B analysis output corresponds to the NCBI taxonomy classification.

**Figure 5 ijms-25-09813-f005:**
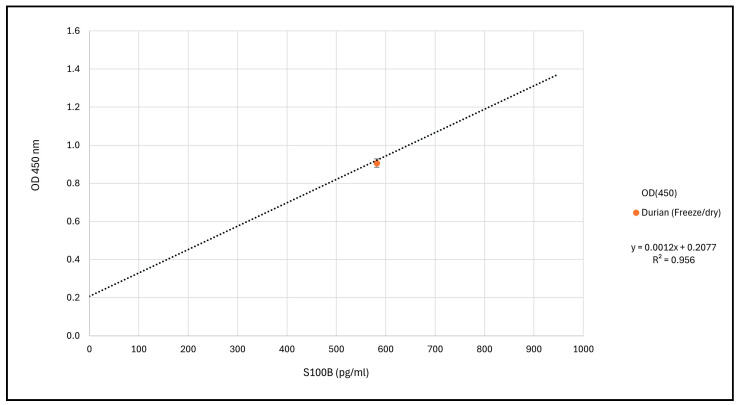
The standard curve of a representative experiment (Abcam, ab234573, Cambridge, UK) and the location of a durian fruit extract regarding the concentration of S100B-like material.

**Table 1 ijms-25-09813-t001:** S100B homologs in plants. The table shows the comparison among the aligned proteins having the higher E-value in the alignment result with the reference structure S100B. Root Mean Square Deviation (RMSD full) and TM-align (TM-align score full, version 2022/04/15) were initially calculated with the software TM-align [22] on the full proteins, with normalization based on the residues of S100B. Solvent Accessible Surface Area (SASA) calculations were performed without normalization, aiming to provide an approximate indicator of the size of the proteins. Afterwards, the conserved N-terminal S100B domain was isolated, and the root mean square deviation (RMSD conserved) and the TM-align score (TM-align score conserved) were calculated for those residues as well.

Name File	Accession	Description	Organism	RMSD	TM-Align Score	SASA [Å^2^]	RMSD Conserved	TM-Align Score Conserved
S100B	P04271	S100B	*Homo Sapiens*			6504.42		
Adansonia digitata	Q6EIJ6	Maturase K	*Artocarpus heterophyllus*	3.67	0.41058	55,361.468	2.04	0.47283
Adiantum capillus-veneris	A0A9D4UGI1	EF-hand domain-containing protein	*Adiantum capillus-veneris*	2.46	0.55553	7227.491	2.32	0.74548
Ceratopteris richardii	A0A8T2T945	EF-hand domain-containing protein	*Ceratopteris richardii*	3.06	0.63275	12,912.797	2.36	0.73105
Cuscuta europaea	A0A9P0ZHW9	EF-hand domain-containing protein	*Cuscuta europaea*	3.8	0.61591	11,235.978	2.85	0.7075
Durio zibethinus	A0A6P6AZ13	Probable calcium-binding protein	*Durio zibethinus*	2.74	0.66936	11,374.781	2.63	0.71475
Gossypium klotzschianum	A0A7J8USE5	EF-hand domain-containing protein	*Gossypium klotzschianum*	2.43	0.62498	11,317.836	2.41	0.70399
Handroanthus impetiginosus	A0A2G9G7M9	EF-hand domain-containing protein	*Handroanthus impetiginosus*	3.07	0.61603	15,050.241	2.19	0.73368
Hibiscus syriacus	A0A6A3BQP9	PfkB-like carbohydrate kinase family protein	*Hibiscus syriacus*	2.82	0.61948	10,389.12	2.13	0.76271
Malus domestica	A0A498KP84	EF-hand domain-containing protein	*Malus domestica*	3.5	0.60179	37,161.187	2.81	0.63253
Musa acuminata	A0A804HRU8	hypothetical protein	*Musa acuminata* subsp. malaccensis	2.53	0.61628	5706.664	2.03	0.77669
Olea europaea	A0A8S0U3M0	Probable calcium-binding CML18	*Olea europaea* subsp. europaea	2.51	0.63074	12,185.494	2.38	0.7467

**Table 2 ijms-25-09813-t002:** Presence or absence of S100B–like immunoreactive material detected through ELISA assay in samples of plant tissue or derivatives. The source, plant’s scientific name and author of the name and the plant family are indicated.

Sample	Scientific Notation	S100B Presence
Açai (powder)	*Euterpe oleracea*, Von Martius, Arecaceae	+
Banana (fresh)	*Musa acuminata*, Colla, Musaceae	−
Banana (powder)	*Musa acuminata*, Colla, Musaceae	−
Baobab (powder)	*Adansonia*, Linnaeus, Malvaceae	+
Broccoli (fresh)	*Brassica oleracea* var. italica, Linnaeus, Brassicaceae	−
Cabbage (fresh)	*Brassica oleracea* var. capitata, Brassicaceae	−
Cocoa (powder)	*Theobroma cacao*, Linneo, Malvaceae	−
Durian (fresh)	*Durio zibethinus*, Linnaeus, Malvaceae	+
Durian (Freeze/dry)	*Durio zibethinus*, Linnaeus, Malvaceae	+
Durian (powder)	*Durio zibethinus*, Linnaeus, Malvaceae	+
Graviola (powder)	*Annona muricata*, Linnaeus, Annonaceae	+
Jack fruit (lyophilized)	*Artocarpus heterophyllus*, Lamarck, Moraceae	+
Kiwi (fresh)	*Actinidia chinensis*, Planch, Actinidiacee	−
Kombucha(powder)	*Camellia sinensis*, Linnaeus, Theaceae	+
Laurel (fresh)	*Laurus nobilis*, Linnaeus, Lauraceae	+
Mela Annurca (*cps*)	*Malus domestica*, Borkhausen, Rosaceae	+
Reishi Mushroom (powder)	*Ganoderma lucidum*, (Curtis) P. Karst. Ganodermataceae	−
Sage (fresh)	*Salvia officinalis*, Linnaeus, Lamiaceae	+
Salad (fresh)	*Lactuca sativa*, Linnaeus, Asteraceae	−
Spinach (fresh)	*Spinacia oleracea*, Linnaeus, Amaranthaceae	−
Spinach (powder)	*Spinacia oleracea*, Linnaeus, Amaranthaceae	−
Sunflower (powder)	*Helianthus annuus*, Linnaeus, Asteraceae	−

## Data Availability

The data that support the findings of this study are available from the corresponding author upon reasonable request.

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
