# Peer review of "In Silico Predicting the Presence of the S100B Motif in Edible Plants and Detecting Its Immunoreactive Materials: Perspectives for Functional Foods, Dietary Supplements and Phytotherapies"

_ijms, 2024, doi:10.3390/ijms25189813_

Round 1

Reviewer 1 Report

Comments and Suggestions for Authors

The manuscript entitled “The multifaceted S100B motif is detectable in edible plants: perspectives for functional foods, dietary supplements and phytotherapies” focuses on an interesting topic.

Particularly, authors attempted to evaluate the presence of S100B motif in plant genomes, and of S100B immunoreactive material in different plants’ extracts, thus, fostering the protein as a dietary supplement or active component in functional foods and a novel principle for phytotherapies.

The aims of the study are within the scope of the journal.

One minor point: Figure 3 and Figure 4 are difficult to read. 

Comments on the Quality of English Language

Minor editing of English language required.

Author Response

Reviewer 1

Comments and Suggestions for Authors

The manuscript entitled “The multifaceted S100B motif is detectable in edible plants: perspectives for functional foods, dietary supplements and phytotherapies” focuses on an interesting topic.

Particularly, authors attempted to evaluate the presence of S100B motif in plant genomes, and of S100B immunoreactive material in different plants’ extracts, thus, fostering the protein as a dietary supplement or active component in functional foods and a novel principle for phytotherapies.

The aims of the study are within the scope of the journal.

One minor point: Figure 3 and Figure 4 are difficult to read.

Comments on the Quality of English Language

Minor editing of English language required.

R: Thank you for your kind comments. The entire manuscript has been critically revised to ameliorate English language and Figures.

Reviewer 2 Report

Comments and Suggestions for Authors

The authors provide an interesting study on S100B and its role in gut microbiota eubiosis. Being secreted in human breast milk may suggests a physiological trophic function in newborn development.

A bioinformatic workflow was used to identify the presence of S100B motif in plants. The immunoreactivity of S100B from vegetables or fruits samples was tested using ELISA 25 assay.

The S100B motif was found in the genome of different edible plants belonging to the Viridiplantae clade, such as Durio zibethinus or Malus domestica and other medicinal species. S100B immunoreactive material was also detected in samples from fruits or leaves. 

The authors conclude that the presence of S100B-like molecules in plants may provide the basis exploring beneficial effects of plants or derivatives containing the principle of Calcium binding proteins.

Comments on the Quality of English Language

minor English edition required

Author Response

Reviewer 2

The authors provide an interesting study on S100B and its role in gut microbiota eubiosis. Being secreted in human breast milk may suggest a physiological trophic function in newborn development. A bioinformatic workflow was used to identify the presence of the S100B motif in plants. The immunoreactivity of S100B from vegetables or fruit samples was tested using ELISA 25 assay. The S100B motif was found in the genome of different edible plants belonging to the Viridiplantae clade, such as Durio zibethinus or Malus domestica and other medicinal species. S100B immunoreactive material was also detected in samples from fruits or leaves.  The authors conclude that the presence of S100B-like molecules in plants may provide the basis exploring beneficial effects of plants or derivatives containing the principle of Calcium binding proteins.

Comments on the Quality of English Language: minor English edition required

R: Thank you for your comments. The entire manuscript has been critically revised to ameliorate English language

Reviewer 3 Report

Comments and Suggestions for Authors

S100B is known to be a Ca2+ binding protein expressed in the brain, skin, breast, and placenta, adipose tissue, kidneys, colon and testicles. The current study conducted an in silico study and ELISA analysis to study the possible presence of the protein in the plant. There are some points that, if adhered to in the manuscript, may result in good research work that opens horizons for several future studies:

From the title of the manuscript to the discussion section, it must be stated that the study is expected, which is inconsistent with the expression “detectable.” This is because the in silico study is merely a prediction and is not an analysis or measurement. Also, the ELISA analysis was not done using a kit specifically designed for plants, but it is more suitable for humans and rodents because it contains their antibodies.

Giving the impression that S100B is beneficial and that plants containing it can represent dietary supplements or functional food due to its presence is not correct and contradicts all previous studies that indicated that S100B is associated with multiple sclerosis, cancer, SARS-CoV-2, acute neural injury, Alzheimer’s and Parkinson’s diseases, amyotrophic lateral sclerosis, schizophrenia, epilepsy, bipolar disorder, depression, obesity, diabetes, inflammatory bowel disease, psoriasis, muscular dystrophy, and retinal disorders. Some of these studies were written by the same authors and were cited in this study, but without addressing the harmful effects of protein. In addition, the study dealt with descriptive rather than quantitative estimation of protein in plants. I think it would be better to direct researchers to study the physiological role of protein in plants and its nutritional metabolism in living organisms.

The manuscript needs careful linguistic review.

Comments on the Quality of English Language

 Minor editing of English language is required.

Author Response

Reviewer 3

S100B is known to be a Ca2+ binding protein expressed in the brain, skin, breast, and placenta, adipose tissue, kidneys, colon and testicles. The current study conducted an in silico study and ELISA analysis to study the possible presence of the protein in the plant. There are some points that, if adhered to in themanuscript, may result in good research work that opens horizons for several future studies: l From the title of the manuscript to the discussion section, it must be stated that the study is expected, which is inconsistent with the expression “detectable.” This is because the in silico study is merely a prediction and is not an analysis or measurement. Also, the ELISA analysis was not done using a kit specifically designed for plants, but it is more suitable for humans and rodents because it contains their antibodies.

R: We would like to express our gratitude to the reviewer for the consideration of this study as a potential source of valuable research that could have the way for further investigations in the future. In light of the suggestions put forth by the reviewer, we have opted to utilize the term 'expected' in lieu of 'detectable' for in silico studies within the manuscript. Additionally, we have amended the title, which is currently 'The multifaceted S100B motif is expected in silico and S100B-like immunoreactive material is detectable in edible plants: perspectives for functional foods, dietary supplements and phytotherapies'. A kit designed for use with humans and rodents was employed, as kits specifically intended for use with plants are not currently available. This study represents the inaugural investigation of this topic. In light of this caveat, the manuscript and title have been amended to use the term "S100B-like immunoreactive material" in place of "S100B-immunoreactive material." This aspect has also been clarified in the revised version of the manuscript's discussion section.

Giving the impression that S100B is beneficial and that plants containing it can represent dietary supplements or functional food due to its presence is not correct and contradicts all previous studies that indicated that S100B is associated with multiple sclerosis, cancer, SARS-CoV-2, acute neural injury, Alzheimer’s and Parkinson’s diseases, amyotrophic lateral sclerosis, schizophrenia, epilepsy, bipolar disorder, depression, obesity, diabetes, inflammatory bowel disease, psoriasis, muscular dystrophy, and retinal disorders. Some of these studies were written by the same authors and were cited in this study, but without addressing the harmful effects of protein. In addition, the study dealt with descriptive rather than quantitative estimation of protein in plants. I think it would be better to direct researchers to study the physiological role of protein in plants and its nutritional metabolism in living organisms.

R: The S100B protein is known to exert a pathogenic effect at high concentration, as correctly reported by the Reviewer, but a trophic beneficial effect at low physiological concentration (for reviews Thelin et al, 2017; Michetti et al, 2023), so that the pathogenic role played by the protein in different pathological conditions, as reported by the Reviewer, in fact does not contradicts the perspective of a beneficial effect of S100B as dietary constituent in plants, and fully fits  the current literature. This consideration, which is in fact indicated in the text (Introduction section and Discussion section), is now more clearly expressed in the present revised version of the manuscript (Discussion section). Indeed, since any quantitative estimation of the S100B-like immunoreactive material in plants would be performed having mammalian purified S100B protein as the standards, in our opinion a quantitative estimation of S100B in plants, although possible of course, would do not make much sense at present knowledge, so that the mere indication of the presence of S100B-like immunoreactive material in plants would offer novel information preparing perspectives of further interest. However, we added Figure 5 showing a representative experiment where the standard curve with the locations of symbols related to some plant samples are indicated, as a representative experiment. We want to thank the Reviewer for the invitation to study physiological role of protein in plants and its nutritional metabolism in living organisms, which will constitute an interesting and promising line of research, also opened by the present study (-Thelin EP, Nelson DW, Bellander BM.Acta Neurochir (Wien). A review of the clinical utility of serum S100B protein levels in the assessment of traumatic brain injury 2017 Feb;159(2):209-225. doi: 10.1007/s00701-016-3046-3; -Michetti F, Clementi ME, Di Liddo R, Valeriani F, Ria F, Rende M, Di Sante G, Romano Spica V.Int J Mol Sci. The S100B Protein: A Multifaceted Pathogenic Factor More Than a Biomarker.2023 May 31;24(11):9605. doi: 10.3390/ijms24119605).

The manuscript needs careful linguistic review.

R: Thanks, the entire manuscript has been critically revised to ameliorate English language.

Round 2

Reviewer 1 Report

Comments and Suggestions for Authors The authors have satisfactorily made the necessary changes to the manuscript. The manuscript is now suitable for publication.

Author Response

Thank you for your time in reviewing this manuscript.

Reviewer 2 Report

Comments and Suggestions for Authors

no comments

Author Response

(The authors gave the same response as above.)

Reviewer 3 Report

Comments and Suggestions for Authors

The authors make the suggested amendments, but I suggest amending the title to: In silico predicting the presence of the S100B motif in edible plants and detecting its immunoreactive materials: perspectives for functional foods, dietary supplements and phytotherapies.

Author Response

Done. The title was amended as suggested.